Colour change of twig-mimicking peppered moth larvae is a continuous reaction norm that increases camouflage against avian predators

Eacock Amy bs0u917b@liverpool.ac.uk amyeacock1@gmail.com 1
Rowland Hannah M. 2
Edmonds Nicola 1
Saccheri Ilik J. 1
1 Institute of Integrative Biology, University of Liverpool , Liverpool , United Kingdom
2 Predators and Prey Research Group, Max Planck Institute for Chemical Ecology , Jena , Germany
Harrison Xavier
Electronic publication date: 2017 Nov 14
Publication date: 2017
Volume: 5
Electronic Location ID: e3999
Received 2017 Jul 28; Accepted 2017 Oct 17
Copyright: ©2017 Eacock et al.
Copyright year: 2017
Copyright holder: Eacock et al.
License: This is an open access article distributed under the terms of the Creative Commons Attribution License, which permits unrestricted use, distribution, reproduction and adaptation in any medium and for any purpose provided that it is properly attributed. For attribution, the original author(s), title, publication source (PeerJ) and either DOI or URL of the article must be cited.
License URL: https://creativecommons.org/licenses/by/4.0/

Keywords: Colour change, Camouflage, Reaction norm, Biston betularia, Masquerade, Polyphenism, Predator-prey interactions

Funding: Natural Environment Research Council NE/J022993 This work was supported by the Natural Environment Research Council (Grant no. NE/J022993). The funders had no role in study design, data collection and analysis, decision to publish, or preparation of the manuscript.

==============================
Camouflage, and in particular background-matching, is one of the most common anti-predator strategies observed in nature. Animals can improve their match to the colour/pattern of their surroundings through background selection, and/or by plastic colour change. Colour change can occur rapidly (a few seconds), or it may be slow, taking hours to days. Many studies have explored the cues and mechanisms behind rapid colour change, but there is a considerable lack of information about slow colour change in the context of predation: the cues that initiate it, and the range of phenotypes that are produced. Here we show that peppered moth (Biston betularia) larvae respond to colour and luminance of the twigs they rest on, and exhibit a continuous reaction norm of phenotypes. When presented with a heterogeneous environment of mixed twig colours, individual larvae specialise crypsis towards one colour rather than developing an intermediate colour. Flexible colour change in this species has likely evolved in association with wind dispersal and polyphagy, which result in caterpillars settling and feeding in a diverse range of visual environments. This is the first example of visually induced slow colour change in Lepidoptera that has been objectively quantified and measured from the visual perspective of natural predators.

Introduction

Some of the most diverse and visually striking phenotypes seen in nature are those of camouflaged animals (Stevens & Merilaita, 2009). Background matching, or crypsis, is a common anti-predator strategy that has provided a test-bed for the theory of evolution through natural selection (Wallace, 1879; Wallace, 1889). Crypsis is selected for by visual predators such as birds (Merilaita, Lyytinen & Mappes, 2001), whereby prey that match the colour/pattern of the surrounding backgrounds survive for longer than non-matching prey (Endler, 1981; Merilaita, Scott-Samuel & Cuthill, 2017). In heterogeneous habitats, comprised of visually contrasting patches, or a gradient from one habitat type to another (Fig. 1), optimising crypsis to all of the background components presents a challenge (Merilaita, Tuomi & Jormalainen, 1999). One solution to this problem is a genetic polymorphism, which can produce two or more morphs that are specialised to different patch types (Merilaita, Lyytinen & Mappes, 2001; Surmacki, Ozarowska-Nowicka & Rosin, 2013). However, a species with a genetically fixed phenotype is restricted to camouflage on one background, or limited camouflage across varied patch colours (Fig. 1A). Therefore, in environments that change appearance across small temporal and spatial scales, detrimental phenotype-environment mismatching can occur (Cook et al., 2012; Farkas et al., 2015). In this case, selection may favour phenotypic plasticity, enabling individuals to actively change their appearance to utilise different habitat patches without compromising camouflage (Fig. 1B; Stevens, 2016). An example of plasticity is colour change, which is a topic of current research interest and can be used to study the adaptive value and the physiology of camouflage (Duarte, Flores & Stevens, 2017).

Figure 1 Possible camouflage strategies of caterpillars in response to visually heterogeneous environments.

(A) In an environment composed of different coloured patches, caterpillars with a fixed genetic phenotype achieve compromised crypsis on all backgrounds. (B) The same habitat scenario as A but with larvae specialised to match one patch type, either by genetic polymorphism, restricting individuals to one patch colour, or by plastic polyphenism, in principle allowing individual larvae to move between patches and switch colour to match their background. (C) Larvae with genetic polymorphism or plastic polyphenism inhabiting a graded environment with intermediate colour patches, where phenotypes match the extreme, but not the intermediate backgrounds. (D) An environmental gradient with intermediate backgrounds, where larvae produce a continuous colour response to background colour, allowing utilisation of each patch colour.

Rapid colour change (<2 h), as reported in fish, cephalopods, and amphibians has been widely studied (Allen et al., 2015; Buresch et al., 2011; Hanlon et al., 2009), and much is known about how chromatophores produce rapid changes in colour and pattern in these systems (Kingston et al., 2015; Mathger & Hanlon, 2007). Comparatively slower colour changes (days to months) occur in some arthropod and fish species (Llandres et al., 2013; Ryer et al., 2008). In many of these systems we still do not know whether slow colour change is adaptive, nor do we know the precise cues or biochemical processes involved. A number of potential cues have been proposed, with dietary and visual cues receiving most attention (Duarte, Flores & Stevens, 2017; Stevens & Merilaita, 2009).

One example of a diet-induced phenotypic switch, or polyphenism, is seen in the larval stage of the moth Nemoria Arizona, which resembles inedible objects in its environment (Greene, 1989). In the spring the larvae resemble oak catkins, and in the summer they look like the branches of oak. This form of visual resemblance to inanimate objects is referred to as masquerade (Skelhorn, Rowland & Ruxton, 2010). Masquerade enables prey to avoid attack because predators misclassify these prey, rather than failing to detect them (Skelhorn et al., 2010b). The larvae of the peppered moth (Biston betularia) also masquerade as the twigs of their foodplant and change colour to match them (Noor, Parnell & Grant, 2008; Poulton, 1892). These brown and green colour morphs occur in response to the background colour on which the larvae rest (Noor, Parnell & Grant, 2008; Poulton, 1892). Changing appearance in response to background cues in the environment may be beneficial for animals that masquerade, as masquerade is often associated with polyphagy (Higginson et al., 2012). Visually hunting predators, like birds, heavily predate caterpillars that do not display warning colours (Lichter-Marck et al., 2015), and twig-mimicking caterpillars that do not match the twigs they rest on are also more likely to be predated (Skelhorn & Ruxton, 2010). Therefore, the ability to change colour could enhance masquerade in the wider range of environments these prey are likely to encounter, and consequently reduce their foraging restrictions (Ruxton, Sherratt & Speed, 2004).

It is important to determine the exact cues eliciting colour change, as these cues initiate the colour change cascade (Duarte, Flores & Stevens, 2017), and can therefore provide information on the evolution of adaptive colour and the mechanisms of colour production (Cuthill et al., 2017). Visual stimuli exist in two forms: achromatic (luminance), and chromatic (hue/chroma). Responses to achromatic stimuli (luminance) have been reported in sand fleas, geckos, toads, and flatfish (Polo-Cavia et al., 2016; Ryer et al., 2008; Stevens et al., 2015; Vroonen et al., 2012). Tree frogs (Hyla japonica) adjust their body colour and luminance, to maximise camouflage against visually heterogeneous backgrounds, although the response to achromatic stimuli was stronger (Choi & Jang, 2014; Kang, Kim & Jang, 2016). Many of these studies propose that colour change in these animals is induced by visual cues, but the visual pathways were not explicitly studied, and additional cues such as temperature or texture were often not controlled (Lin, Lin & Huang, 2009; Polo-Cavia et al., 2016; Yamasaki, Shimizu & Fujisaki, 2009).

To address this topic, we conducted a series of experiments to explore the type of visual cues that elicit colour change in B. betularia. The colour change in B. betularia has previously been described as a polyphenism: a switch of phenotype (Noor, Parnell & Grant, 2008). However, in the only study so far to investigate this behaviour, Noor, Parnell & Grant (2008) only provided two discrete stimuli: green vs. brown, and measured colour subjectively from a human perspective. The larvae of B. betularia are polyphagous and wind dispersed at first instar (Noor, Parnell & Grant, 2008; Tietz, 1972). The wide variety of twig colours between and within host plant species (Edmonds, 2010) presents a highly heterogeneous resting background. Therefore, it may be beneficial for individuals to change appearance on a continuous scale over time (Fig. 1D), known as a reaction norm (Woltereck, 1909). Colour reaction norms have been reported in squid, geckos, and anurans (Kang, Kim & Jang, 2016; Mathger & Hanlon, 2007; Vroonen et al., 2012), and are commonly induced by visual stimuli aquired by the animal about its environment. Reaction norms have not yet been investigated in lepidopteran larvae in this context.

We used calibrated stimuli in order to investigate the adaptive significance of colour change in B. betularia (Stevens & Merilaita, 2009). We manipulated luminance (brightness) and colour, and evaluated the degree to which B. betularia caterpillars are able to respond to intermediate strength cues (i.e., discrete polyphenism vs. reaction norm). We also measured the response to heterogeneous twig colour environments. For the purpose of these experiments, ‘colour’ encompasses hue and chroma. Hue is defined as the direction of the colour vector, and chroma as how different a colour is from achromatic white/black (Stoddard & Prum, 2008). ‘Luminance’ is defined as achromatic intensity, or perceived brightness (Stevens, Lown & Denton, 2014; Stoddard & Prum, 2008). We modelled colour using the avian visual system which allows a more direct adaptive interpretation of larval colour change in B. betularia, compared to using human vision. We tested the following predictions: (1) larvae respond to both colour and luminance; (2) larvae produce intermediate phenotypes in response to changing colour and/or luminance on a continuous scale (i.e., a reaction norm rather than a polyphenism, as suggested by Noor, Parnell & Grant (2008)); (3) when faced with a heterogeneous background, larvae adopt an intermediate colour reflecting the relative proportion of twig colours.

Figure 2 Dowels used for luminance, colour, and heterogeneous environment experiments.

(A–L) represent IB, IG, Bl, BW1, BW2, BW3, Wh, Br, BG1, BG2, BG3, and Gr, respectively.

Materials & Methods

Experimental animals and rearing

All larvae for the various experiments were the F1 offspring from crosses between wild-caught or captively reared adults. Larvae typically undergo five moults, resulting in six instars until pupation (Noor, Parnell & Grant, 2008). In the first instar, larvae appear countershaded, and in subsequent instars the larvae develop colours that appear to match the twigs of the different host plants on which they may rest. The adults occur as a series of more or less discrete morphs differing in the degree of melanism: typica, insularia and carbonaria (Cook & Muggleton, 2003). As previous observations suggested no effect of adult morph on larval colour, or vice versa, some families used in these experiments were segregating for adult melanism alleles, whilst others were fixed for the typica allele (Table S1). For experiments requiring more larvae than one cross could provide, larvae from multiple crosses were split across treatments to minimise any family effects (Table S1). Larvae were initially reared from eggs on intact goat willow (Salix caprea) branches with leaves until second instar. Groups of 25 caterpillars from the same family (full siblings) were then introduced into transparent plastic boxes measuring 279 × 159 × 102 mm (length × width × depth), containing an irregular lattice of twenty 12 cm-long (ten 3 mm and ten 5 mm diameter) painted softwood dowels. The dowels were held in place with a chicken wire mesh frame painted the same colour as the dowels (Fig. 2). All paints used for dowels were from the Dulux Matte range (Table S1). To facilitate cleaning, the base of each box was lined with a plain blue C-Fold 1-ply paper towel and larvae were fed on stripped, stalkless leaves of goat willow (Salix caprea), which was replenished so that the larvae had a constant supply of food. Boxes were regularly cleaned and hands and equipment were washed in dilute bleach (10%) between handling of boxes to reduce risk of disease transmission. Experiments were conducted in a Sanyo Versatile Environment Test Chamber (MLR-351), with light intensity set at 15,000 lx during the day. Boxes were arranged two on each shelf, 20 cm apart, leaving a 60 cm height space with a shelf between boxes.

Colour and luminance quantification

Spectrophotometric analysis

Reflectance measurements of larvae and dowels were taken using an Ocean optics USB2000 spectrophotometer, with specimens illuminated at 45° to normal by a DH1000 balanced halogen deuterium light source. The measuring spot diameter was 3 mm, with spectra recorded at 0.34 nm intervals from 300 to 700 nm and measured relative to a WS-1 reflectance standard. Spectrophotometry data were visualised using Overture (v.1.0.1). Reflectance spectra were reduced to 1 nm intervals within the 300–700 nm range using customised code (provided by I Cuthill).

Firstly, to determine differences in ‘colour’ between larvae and the dowels from the colour experiment, the predicted photon catches of cone types (longwave LW, mediumwave MW, shortwave SW, ultraviolet UV, and double dorsal DD) of a blue tit, Cyanistes caeruleus, were modelled for each spectrum in tetrahedral colour space following the Vorobyev & Osorio (1998) model, using a program written in MATLAB (provided by I. C. Cuthill). We used the blue tit to represent the avian visual system because there is good visual data available and this species is likely a natural predator of B. betularia larvae. Cone stimulation values were converted to Cartesian coordinates and plotted in a tetrahedral space using a MATLAB program (Stoddard & Prum, 2008), such that each cone is represented by an axis. This colour space is useful because if a colour stimulates only one cone type, then its coordinates lie at the appropriate tip of the tetrahedron, and when all four cone types are equally stimulated the point lies at the origin. To provide a simpler measure of colour, we calculated greenness as the ratios between the cone catch values of the mediumwave and longwave photoreceptors [MW/(MW + LW)], which represent opponent mechanisms, following Arenas & Stevens (2017). For the achromatic dowel experiment we created a stimulus that increased in luminance in the absence of ‘colour’ (black to white), therefore we did not model response to colour, only luminance. We analysed only the blue tit double dorsal cone catch, as these cones mediate luminance vision (Campenhausen & Kirschfeld, 1998; Osorio & Vorobyev, 2005).

We modelled the ease with which an avian predator might discriminate between dowels and larvae using just noticeable differences (JND; see Vorobyev & Osorio, 1998 for equations). For chromatic contrasts, we used spectral sensitivities of the blue tit using relative cone ratios of SW = 0.7111; MW = 0.9926; LW = 1.0 and UV = 0.3704 (Hart et al., 2000), with a Weber fraction of 0.05 and idealized irradiance (D65). To model luminance JNDs, we used blue tit double dorsal (DD) cones. JND <1.00 indicate that two stimuli are indiscriminable; stimuli differing by 1–3 JND units are only discriminable under good viewing conditions; and stimuli showing values above this should be distinguishable with increasing ease (Stevens et al., 2015).

Photographic analysis

Colour/luminance analysis on larvae from the heterogeneous dowel experiments was performed using calibrated photographs, as the spectrophotometer was not available when these experiments were conducted. Photographs of individual larvae were normalised to a standardised grey background (18%) and linearised to 32-bit files using the Image Calibration and Analysis Toolbox (Troscianko & Stevens, 2015) in ImageJ (v.1.49p). RGB values were extracted from processed images using ImageJ (Schneider, Rasband & Eliceiri, 2012), from an average of six dorso-lateral measurements per larva: one from each side of the 3rd thoracic, and 2nd and 6th abdominal segments. An average measure of percentage greenness was then calculated across the six measurements using G/(R + G + B) * 100 from RGB ratios. Although objective, these measurements were not modelled using an avian visual system.

Experimental treatments

A total of four experiments were conducted to test three main hypotheses concerning the nature of the environmental cue and the phenotypic response (Table 1).

Table 1 Summary of experiments and hypotheses.

Experiment	Twig environment	Hypothesis	
i	Contrasting colour	1a. Larvae respond to differences in twig colour	
ii	Luminance gradient	1b. Larvae respond to differences in twig luminance
2a. Larvae can produce intermediate responses to twig luminance	
iii	Colour and luminance gradient	2b. Larvae can produce intermediate responses to twig colour and luminance	
iv	Heterogeneous environment	3. Larvae generalise across twig colours	

(i) Colour treatments

Dowels were painted either isoluminant green or isoluminant brown (Fig. 2: IG, IB) to create two treatments that differed in overall colour (colour JND: 21.2) and greenness (Two sample t-test, t9.99 =  − 16.86, P < 0.0001), but not luminance (luminance JND: 1.8, Two sample t-test, t5.86 = 0.37603, P = 0.7201). Five replicate boxes were used for each treatment (Table S1). Larvae were reared on a 12:12 hour day: night cycle, at 24 °C in the day and 18 °C at night. Once larvae had reached final instar, six reflectance measurements per larva were taken with a spectrophotometer, three from each lateral surface, on the 3rd thoracic, and 2nd and 6th abdominal segments. These segments were chosen to obtain repeated measurements of the main body colour, excluding any prominent markings.

(ii) Luminance gradient

Five luminance treatments were created using painted dowels (Fig. 2: B1, BW1, BW2, BW3, Wh) increasing in luminance (ANOVA, F4 = 8,415, P < 0.0001) from near-black to white (Fig. S1A, Table S1), and approximately equal in colour. Paint was matched to the grey standards from a Gretag Macbeth colour chart using a Gretag Macbeth colour scanner at a UK hardware store (B&Q). Three replicate boxes per treatment were used (Table S1). Larvae were reared on a 15:9 hour day: night cycle at 21 °C in the day and 19 °C at night. Once larvae had reached final instar, four reflectance measurements were taken with a spectrophotometer from the dorsal surface of each caterpillar, on the 3rd thoracic segment, and the 2nd, 4th and 6th abdominal segments.

(iii) Colour and luminance gradient

We mixed brown (Br) and green (Gr) paint in three different ratios to give a total of five treatments that ranged from brown to green (Table S1 , Fig. 2: Br, BG1, BG2, BG3, Gr). These treatments differed in greenness (Fig. S1D; ANOVA, F4 = 1,378, P < 0.0001) and luminance (Fig. S1C; ANOVA, F4 = 82.68, P < 0.0001), although greenness of BG1, BG2 and BG3 was less than expected based on the proportion of Gr paint in the mixture. Three replicate boxes per treatment were used. Larvae were reared on a 12:12 hour day: night cycle, at 24 °C in the day and 18 °C at night. Once larvae had reached final instar, six reflectance measurements per larva were taken as for colour treatments.

(iv) Heterogeneous dowel environment

Five treatments were created using only two colours of dowel, brown and green (Fig. 2: Br, Gr), but in different ratios: 100% brown, 70 brown: 30 green, 50 brown: 50: green, 30 brown: 70 green, and 100% green (Table S1). Larvae were reared on a 15:9 hour day: night cycle at 21 °C in the day and 19 °C at night. Once final instar was reached, the dorsal surface of each larva was photographed on a standard grey card background using a Nikon D80 digital camera, 60 mm macro lens with the following settings: 1/60 s (shutter speed), 16 (F), 400 (ISO), cloudy (white balance), 2× Nikon Speedlight SB-400 External Flash.

Statistical analyses

All statistical analyses were performed using R version 3.1.0 (R Core Team, 2014). Responses to contrasting colour cues were compared using a linear mixed model in the lme4 package in R (Bates et al., 2015), with replicate nested within treatment. Luminance gradient and greenness gradient response means for each of the five treatment levels were compared using a one-way ANOVA. Polynomial models from orders 1–4 were fitted to the luminance and greenness correlations to explore the relationship between environmental gradient and larval response. All polynomial models are presented as fitted in Fig. S2. R2 value, visual fit, and examination of plotted residuals were used to determine the best model for each correlation. ANOVA was performed to look for significant differences in fit between models. The ‘greenness’ response of larvae reared under different degrees of dowel-colour heterogeneity was analysed by comparing means using a one-way ANOVA. Homogeneity of variance between treatment medians was explored using Levene’s test.

Figure 3 The response of B. betularia larvae to a difference in dowel colour.

(A) Representative final instar B. betularia larvae from each isoluminant treatment resting on their corresponding dowel. (B) The average position of final instar B. betularia larvae and their corresponding dowels within the ultraviolet-sensitive (UVS) avian tetrahedral colour space when viewed by a blue tit, Cyanistes caeruleus, under bright daylight conditions. Asterisks represent dowels, rhombuses represent larvae, from brown and green treatments, respectively. The plot illustrates the stimulation of the short (S), medium (M), long (L), and UV (U/V) wavelength-sensitive photoreceptors and is shown from the MW–LW plane. (C) Greenness as perceived by a blue tit under bright daylight conditions of final instar B. betularia larvae reared under isoluminant dowel treatments, where IBL, isoluminant brown larvae and IGL, isoluminant green larvae. The numbers following the letters indicate replicate boxes within each treatment. IBD, isoluminant brown dowel and IGD, isoluminant green dowel. Photo credit: Arjèn Van’t Hof.

Results

Response to colour (isoluminant dowels)

Larvae presented with the isoluminant green (IG) treatment were significantly greener than those in the isoluminant brown (IB) treatment (Fig. 3A; F11, 125 = 33.69, P < 0.0001). In colour space, the larvae resembled their own treatment colour more closely than the alternative treatment colour (Fig. 3B), and green and brown larvae were discriminable to a bird (colour JND: 11.3). The average response was consistent across replicates within treatments (Fig. 3C), but the discrepancy between larval and dowel greenness was greater for larvae reared on green dowels (colour JND: 9.9), than those reared on brown dowels (colour JND: 8.3). Colour change took approximately 14–21 days to complete, depending on the individual. Although colour change is not restricted to the final instar, to the human eye, noticeable change did not occur until 4th instar (Edmonds, 2010).

Response to luminance gradient

Larvae responded to dowel luminance (F4 = 148.2, P < 0.0001), ranging from very pale on white dowels to very dark on black dowels, with intermediate degrees of luminance on grey dowels (Fig. 4A). The relationship between larvae and dowel luminance was significantly cubic (F3, 261 = 156.3, P < 0.0001, R2 = 0.64): relatively steep at the extremes and shallow at intermediate luminance (Fig. 4B). This shape was due to smaller luminance differences between larvae from intermediate treatments (BW1 vs. BW2: luminance JND: 1.2; BW2 vs. BW3 luminance JND 3.4). The differences between larvae from the two extremes of the gradient (black and white) vs. intermediate were larger (Bl vs. BW1, luminance JND: 17.4; and Wh vs. BW3, luminance JND: 13.2).

Figure 4 The response of B. betularia larvae to a gradient in dowel luminance.

(A) Representative final instar B. betularia larvae from each luminance treatment resting on their corresponding dowel. Dowel treatments shown from left to right: Black (Bl), Dark grey (BW1), Mid grey (BW2), Light grey (BW3), White (Wh). (B) Average luminance of final instar B. betularia larvae reared under the five luminance treatments, as perceived by a blue tit (Cyanistes caeruleus) under bright daylight conditions. Solid line is the fitted cubic polynomial; dotted and dashed lines, provided for comparison, represent the linear (idealised continuous reaction norm) and stepped (two-state polyphenism) responses, respectively. Photo credit: Arjèn Van’t Hof.

When comparing larvae to their corresponding dowels, larvae from higher luminance treatments (BW2, BW3 and Wh) were most different from their dowels (luminance JNDs: 35.9, 43.4 and 35.8, respectively). Black (Bl) and dark grey (BW1) treatments showed comparatively lower JNDs between larvae and dowels (luminance JNDs: 28.9 and 20.0, respectively).

Response to colour and luminance gradient

B. betularia larvae adjusted both greenness (F4 = 120.6, P < 0.0001) and luminance (F4 = 82.68, P < 0.0001) in response to dowel stimuli (Fig. 5A), showing a significant positive quadratic correlation between larvae and dowel greenness (Fig. 5B; F2, 277 = 225.6, R2 = 0.62, P < 0.0001). Two of the intermediate brown-green treatments (BG1 and BG2) were very close in greenness (Fig. 5B), and discrimination between them was low (colour JND: 4.61, luminance JND: 2.9). Larvae from these treatments followed this pattern closely, with identical greenness of 0.45 (Fig. 5B) and low discrimination values (colour JND: 2.4, luminance JND: 2.9). The colour discrepancy between larvae and dowels from the brown treatment (Br) was smaller (colour JND: 5.8) than for the green (Gr) treatment (colour JND: 14.1).

Figure 5 The response of B. betularia larvae to a gradient in dowel colour and luminance.

(A) Photograph of final instar B. betularia larvae from each colour treatment resting on their corresponding dowel. Dowel treatments shown from left to right: Brown (Br), More brown (BG1), Brown-green (BG2), More green (BG3), Green (Gr). (B) Average greenness of dowels vs. B. betularia larvae exposed to dowels from each of the five treatment groups, as perceived by a blue tit (Cyanistes caeruleus) under bright daylight conditions. Solid line is the fitted quadratic polynomial; dotted and dashed lines, provided for comparison, represent the linear (idealised continuous reaction norm) and stepped (two-state polyphenism) responses, respectively. Photo credit: Lukasz Lukomski.

Response to heterogeneous colour environments

As the proportion of green dowels relative to brown dowels in each enclosure increased, the average greenness of B. betularia larvae in each enclosure also increased (Fig. 6; F4 = 16.2, P < 0.0001). Between-individual variance in larval greenness was significantly higher in the three heterogeneous than in the two homogeneous colour environments (Levene’s test, F4 = 16.558, P < 0.0001). This result still held when the most variable treatment was removed (Levene’s test, F3 = 8.3093, P < 0.0001). The apparent change in the average level of greenness in mixed treatments (Fig. 6) arose predominantly from changes to the ratio of ‘green’: ‘brown’ individuals, which was roughly in line with the dowel ratios, rather than every larva taking on an intermediate colour (Fig. S3).

Figure 6 The response of B. betularia larvae to different ratios of green and brown dowels.

Percentage of green dowels in each treatment vs. the percentage of greenness of B. betularia larva as calculated by RGB analysis.

Discussion

Biston betularia caterpillars changed colour to match the twigs upon which they rested, across all four experiments. Our results support the prediction that larvae would be able to respond to both colour and luminance (hypotheses 1a and 1b in Table 1). Larvae reared on green and brown dowels changed colour to match those dowels, and larvae reared on black and white dowels changed their luminance. Many other species can adjust luminance to enhance their camouflage from predators, such as flounders, sand fleas, and toads (Fairchild & Howell, 2004; Polo-Cavia et al., 2016; Stevens et al., 2015). Frogs, gobies and cephalopods can rapidly adjust colour in response to visual backgrounds using chromatophores (Hanlon et al., 2009; Kang, Kim & Jang, 2016; Mathger & Hanlon, 2007; Stevens, Lown & Denton, 2014). To our knowledge, our results are the first to show that lepidopteran larvae respond to both colour and luminance, and are likely to use dermal pigments as opposed to chromatophores to change their appearance. We also found that when B. betularia larvae were presented with colour and luminance gradients, the larvae produced intermediate phenotypes, on a continuous scale, to approximately match each background. This supports our second prediction (hypotheses 2a and 2b, Table 1). Intermediate phenotypes have been reported in amphibians, fish, and a number of benthic invertebrates, but the cues and mechanisms controlling these phenotypes have not been rigorously explored (De Bruyn & Gosselin, 2014; Kang, Kim & Jang, 2016; Lin, Lin & Huang, 2009; Skold, Aspengren & Wallin, 2013). Contrary to our third hypothesis (Table 1), larvae did show an increase in average greenness across the treatments with a heterogeneous background. However, this was largely due to an increasing proportion of green individuals compared with brown individuals, rather than every larva becoming greener. Our four experiments provide the first conclusive evidence of intermediate colour change in lepidopteran larvae in response to visually graded or heterogeneous cues. Our results extend our understanding of the type of visual cues that B. betularia larvae use for colour change, and the range of colours they can produce.

Visual control of colour change is well known in animals that exhibit rapid colour change, such as flatfish (Kelman, Tiptus & Osorio, 2006) and cephalopods (Mathger & Hanlon, 2007; Ramirez & Oakley, 2015), where chromatophores, under direct control from the visual nerve system, are responsible for the rapid colour change (Kingston et al., 2015; Messenger, 2001). Other species show comparatively slower responses to background manipulation. For example, shore crabs respond predominantly to luminance cues over colour to match their background (Stevens, Lown & Wood, 2014), and sand fleas are able to match changes in luminance and colour to avoid predation (Stevens et al., 2015). In Lepidoptera, early experimental evidence indicated that colour change was associated with larvae and pupae sensing their visual environment (Poulton, 1890). Since this pioneering work, the evidence collected in support of visually induced colour change in Lepidoptera has been limited and inconclusive: the experiments in B. betularia larvae (Noor, Parnell & Grant, 2008), and two species of hawkmoth larvae, Smerinthus ocellata and Laothoe populi (Grayson & Edmunds, 1989) did not measure colour objectively from the perspective of an ecologically relevant predator, and other potential cues were not controlled for.

In showing that background colour induces the phenotypic change in B. betularia, our results are in accordance with some of the conclusions drawn by Noor, Parnell & Grant (2008). By keeping dietary and tactile cues constant, we also found that B. betularia larvae use visual cues to change colour. However, our results differ from Noor, Parnell & Grant (2008) in that we have found that the response is a continuous reaction norm, not a polyphenism of only two phenotypes. This may be because the experiments by Noor, Parnell & Grant (2008) did not provide a spectrum of background colours, or because the responses of the larvae in Noor, Parnell & Grant (2008) were measured subjectively by assigning individual caterpillars as “best fits” to one of four colour categories. This necessarily reduces any variability to four levels. We objectively measured the colour of both the stimuli and the larvae from the perspective of avian predators. This information provides us with a better understanding of how the highly polyphagous larvae of B. betularia might avoid predation in a changing environment. The range of colour phenotypes that the larvae are able to produce could enable crypsis on a wide range of host plants, reducing costs of lost foraging opportunity, and explaining the higher probability of polyphagy by masquerading species (Ruxton, Sherratt & Speed, 2004). The background matching ability of B. betularia larvae is also likely to allow them to adapt to the blackening of trees and shrubs caused by atmospheric coal pollution. There is no direct evidence for this in B. betularia, as its larvae are very difficult to collect from the wild and occur at low densities. However, the twig-mimicking caterpillars of another geometrid, Odontopera bidentata, which are normally pale to medium brown, or with green (‘lichen’) patches, were uniformly black in the heavily polluted inner-city areas of 1970’s Manchester (Bishop & Cook, 1980).

The continuous relationship we observed between dowel colour and larval colour is non-linear, whereas the standard reaction norm is generally depicted as a linear relationship (Oomen & Hutchings, 2015). Non-linear reaction norms are common in nature; for example, in response to temperature: life history in butterflies (Brakefield, Kesbeke & Koch, 1998), pigmentation in fruit flies (Rocha, Medeiros & Klaczko, 2009), and morphology in sticklebacks (Ramler et al., 2014). The reason that we observed a non-linear relationship may be because colour change is costly (Polo-Cavia & Gomez-Mestre, 2017). However, the cost of colour change and the trade-off between these and foraging costs has yet to be explored in this species. An alternative explanation is that our stimuli did not surpass the thresholds needed to elicit the cascade from vision to colour (Burtt, 1951).

Vision in lepidopteran larvae has been much less studied than the compound eye of the adult stage (Briscoe & Bernard, 2005; Liu et al., 2017; Xu et al., 2013), but it is assumed that the simple ring of eyes or ocelli provides relatively poor vision (Ichikawa, 1990; Lin, Hwang & Tung, 2002). Our results show that B. betularia larvae can perceive differences in brightness and colour, and support the idea that visually induced plastic colour change in arthropods is mediated through the eyes. There is also growing evidence that camouflage may be partly guided by light-detecting opsin proteins outside the eye (Stevens, 2016). Further exploration of the visual processes and pathways that determine the sensitivity and range of colour change is important for understanding camouflage, and has been highlighted as a key area for future work (Duarte, Flores & Stevens, 2017).

In our experiments, there was variation in the degree of colour and luminance matching of the larvae to the dowels. For example, larvae were able to increase luminance as dowel luminance increased, but were always less bright than their corresponding dowels and in some cases would be detectable by birds. Larvae were also not able to closely match the green (Gr) dowel, and again would be detectable by birds. However, we know that resembling twigs is sufficient for masquerade to reduce predation risk, and a lack of perfect colour/luminance match is less detrimental for masqueraders than for cryptic prey (Skelhorn et al., 2010b). This lack of perfect resemblance could be due to physiological constraints, as the white dowels were highly luminant and the green dowels had a very high greenness score. The colours we used were chosen to test the range of colours that B. betularia larvae were able to match, rather than closely resembling the colour/luminance of twigs that individuals are exposed to in nature.

The physiological mechanism behind the colour change in B. betularia is unknown, though preliminary investigations have revealed that the external colour is achieved by varying pigmentation in three primary layers of epidermal tissue (Fig. S4). Cuticular pigments are responsible for colour patterns and have been described in other lepidopteran larvae (Dahlman, 1969; Goodwin, 1953). It is assumed that sequestering bright white or green pigments through a plant diet may be difficult, but yellow and white colouration is achieved with ommochrome pigments in the crab spider, Misumena vatia (Insausti & Casas, 2008). Material properties may also affect luminance, as different materials reflect different amounts of light, which may be the reason for B. betularia larvae achieving lower luminance than dowels. Another explanation for the larvae not achieving a perfect match to their backgrounds is similar to the ideas on imperfect mimicry (Greene & McDiarmid, 1981; Pekar & Jarab, 2011). Masquerade alone enables larvae like B. betularia to avoid being eaten by birds (Skelhorn et al., 2010a), therefore, if there is weak selection against imperfect mimics, then imperfect colour and pattern may not entirely negate the deceptive effect. Kallima butterflies masquerade as dead leaves, closely matching the shape, texture, and colour of the leaf (Suzuki, Tomita & Sezutsu, 2014). It is not currently known whether the shape (and posture), or colour is more important in remaining inconspicuous to predators, but it is thought that relaxed selection on close colour mimicry may occur because cognitive processes of predators (learned discrimination) are more important than sensory processing for visual detection of prey (Stoddard, 2012). Relaxed selection has been suggested as a precursor to phenotypic plasticity (Hunt et al., 2011), and relaxed colour selection in B. betularia could have contributed to colour plasticity in the larvae. However, more research is needed on this topic to understand the origins of colour plasticity in antipredator defences.

The increased variance among individuals produced by the heterogeneous environment treatment implies that the change in mean colour was mostly due to altered ratios of green and brown larvae, which may also be appreciated by inspection of the individual photographs (Fig. S3). This outcome is in contrast to theoretical models on camouflage in heterogeneous environments, which indicate that animals utilising resources on heterogeneous backgrounds should combine markings from each background, instead of optimising camouflage for a single background type (Merilaita, Tuomi & Jormalainen, 1999). This model was tested empirically with bird predators, and in this case prey with colour compromised between two habitats were predated less than those with matching colouration (Merilaita, Lyytinen & Mappes, 2001). However, in certain situations, such as large separation between patch types, specialisation towards one background type is predicted (Houston, Stevens & Cuthill, 2007), and has been observed in animals with fixed (Merilaita, Lyytinen & Mappes, 2001; Pellissier et al., 2011; Sandoval, 1994) and plastic phenotypes (Magellan & Swartz, 2013; Wente & Phillips, 2003). Although patch types (different coloured twigs) within our experimental environments were in close proximity and within easy range of every larva, individual larvae likely receive the strongest cues from the dowel they are resting on and, as they develop, may increasingly choose to rest on that type, reinforcing the specialisation response. Tracking resting behaviour of individual larvae and modelling this colour response using an avian visual system would allow us to make a more solid conclusion. Specialising crypsis to one colour would be a particular benefit to B. betularia larvae, which feed at night and are almost motionless during daylight hours when visual hunters are active. A sedentary lifestyle may accommodate a slow rate of colour change, as also observed in seahorses (Lin, Lin & Huang, 2009), whereas rapid colour change is required for camouflage success if an animal moves rapidly over spatially heterogeneous environments, as observed in fish and cephalopods (Mathger & Hanlon, 2007; Watson, Siemann & Hanlon, 2014).

Conclusions

We show that larvae of the peppered moth use visual cues to closely match the colour and luminance of their background and that this is a continuous response, or reaction norm. The adult and larval stages of B. betularia show alternative evolutionary routes to crypsis, with colour polymorphism under genetic control in the adult moths and reaction norm in the larvae. Both routes achieve protection against avian predation, and it is likely that these contrasting evolutionary strategies have been influenced by differences in life history traits, such as dispersal, reproduction, and feeding behaviour between adults and larvae, as well as physiology. Our results show a novel response in a species belonging to a group of animals whose camouflage potential has been poorly studied in comparison to other taxonomic groups.

Supplemental Information

Figure S1 Distribution of dowel luminance and greenness

(A) Distribution of dowel luminance and greenness. Luminance of dowels used in luminance gradient experiment; (B) Luminance and greenness of isoluminant brown (IBD) and isoluminant green (IGD) dowels; (C) Luminance of dowels used in colour and luminance gradient experiment; (D) Greenness of dowels used in colour and luminance gradient experiment. For explanation of treatment codes see Table S1.

Click here for additional data file.

Figure S2 Polynomial model fitting of larvae luminance and greenness in response to dowel gradient treatments

(A) Scatterplot of dowel and larvae luminance from luminance experiment. (B) Scatterplot of dowel and larvae greenness from colour experiment. Polynomial models represented in both panels (A) and (B) by colours: 1st order (red), 2nd order (green), 3rd order (blue), 4th order (purple).

Click here for additional data file.

Figure S3 Photographs of final instar B. betularia larvae from heterogeneous environment treatments

Photographs of the final instar larvae under treatment 0G (0% Green, 100% brown dowel proportions).

Click here for additional data file.

Figure S4 The external colour of B. betularia larvae is achieved by a three-layer palette

External dorsal surface of green (A) and brown (B) larvae. Dissection of the same larvae reveals that the primary colour in green phenotypes comes from underlying fatty tissue visible through translucent cuticular and epidermal layers (C). In brown phenotypes, there is less green tissue, the epidermis is reddish brown, and the cuticle has pronounced black spots (D).

Click here for additional data file.

Table S1 Experimental design summary

Click here for additional data file.

We would like to thank the reviewers, Tim Caro and Jolyon Troscianko, for their constructive feedback on the manuscript.

Additional Information and Declarations

Competing Interests

Author Contributions

Data Availability

The authors declare there are no competing interests.

Amy Eacock conceived and designed the experiments, performed the experiments, analyzed the data, contributed reagents/materials/analysis tools, wrote the paper, prepared figures and/or tables, reviewed drafts of the paper.

Hannah M. Rowland and Ilik J. Saccheri conceived and designed the experiments, contributed reagents/materials/analysis tools, wrote the paper, reviewed drafts of the paper.

Nicola Edmonds conceived and designed the experiments, performed the experiments, analyzed the data, contributed reagents/materials/analysis tools, reviewed drafts of the paper.

The following information was supplied regarding data availability:

Eacock, Amy (2017): RGB raw data. figshare. https://doi.org/10.6084/m9.figshare.5239657.v2

Eacock, Amy (2017): Raw spectrophotometer files for luminance gradient experiment. figshare. https://doi.org/10.6084/m9.figshare.4780519.v3

Eacock, Amy (2017): Raw spectrophotometer data for chroma gradient experiments. figshare. https://doi.org/10.6084/m9.figshare.4780516.v2

The MATLAB code used for analyzing spectrophotometry data in the article has not yet been published but can be made available upon request from Innes Cuthill at i.cuthill@bristol.ac.uk. The analysis can be repeated by using calculations from Vorobyev & Osorio (1998) with an R package such as Pavo (Maia et al., 2013).

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
