# Peer review of "Colour change of twig-mimicking peppered moth larvae is a continuous reaction norm that increases camouflage against avian predators"

_PeerJ, doi:10.7717/peerj.3999_

## Round 0.1 · original submission · Minor Revisions

· Academic Editor

Minor Revisions

This is a well written paper with some clear results. I particularly like the inclusion of example caterpillar images in relevant figures, which really help to contextualise the results for the reader.

From my own reading of the paper, I would suggest that the introductions be tightened up slightly. While you've done a commendable job of covering a broad range of literature on the background to the study, it tends to read more like a literature review of the taxonomic spread of background-matching studies. You do note the need for direct study of slow colour change, but the impact of this rationale gets slightly lost in the overly-long introduction. Likewise the taxonomic repetition at the start of the discussion, in my opinion, gets in the way of exploring the novelty of the results.

You will note that reviewer 2 has come to the same conclusion. I also agree with reviewer 2 that some more natural history would really help interpretation of the results and contextualise why these animals exhibit the observed camouflage dynamics. Some small changes here will make an excellent manuscript even better.

Reviewer 1 has provided a thoughtful list of statistical tweaks and queries that you may find useful, and which should be addressed.

Other Comments:
Methods: Statistical analyses: There are no citations for R software nor the lme4 package.

Fig. 4 Cubic relationship between dowel luminance and larval luminance
The area under the curve (i.e. polynomial line relative to the idealised continuous reaction norm) is larger at low, compared to high, levels of luminance. Does this mean that variance / error is not constant at all values of x? And does this mean it’s ‘harder’ for the larvae to match lower luminance values ‘exactly’?

·

Basic reporting

The paper is well written and covers the relevant literature.

Experimental design

The experimental design is suitable, though as mentioned below, the authors should justify the switch from spectroscopy to objective imaging (without animal-vision) for the heterogeneity experiment.

The authors use the term “chroma” interchangeably with hue without justifying why. In any case, this should be changed or clarified, and will not affect the overall interpretation.

Validity of the findings

The findings are valid, though there are some areas that require clarification, or could be improved:

A tweak in the “greenness” calculation used might improve the findings and reduce statistical dispersion issues (see general comments).

I think the authors should be much more careful when asserting that the larvae were not adopting an intermediate colour in the heterogeneous treatments as they are unable to prove that the background is heterogeneous from the larval perspective instead of plausible mechanistic or behavioural alternatives.

Statistics: using ordinal statistics might offer a more clean way to analyse the result of different treatment levels (e.g. cumulative link models using the ‘ordinal’ package in R allows for mixed models), though I think the author’s approach is ok. As mentioned above though, the ‘greenness’ measure used creates a large skew in the dataset, which means the green treatment will have a disproportionally more powerful influence on the fitted models than the other treatments. In practice the author’s findings are very clear in that there are intermediate levels of greenness and luminance on the different treatments.

Additional comments

Why was photography used for the heterogeneous treatments, while spectrometry was used for all other treatments? This needs justifying. The images were not converted to bluetit cone-catch quanta images, so will be different to the spectrometer-based measures and the authors should make this more clear to the reader.

The use of imaging means the authors could also look at within-individual heterogeneity. e.g. rather than measure average larval colour, they could see whether larvae on heterogeneous backgrounds also have greater variation. To do this the authors could divide the G channel by the sum of RGB channels, then measure the standard deviation in pixel levels. Higher standard deviations would indicate higher within-individual variation in green and brown patches. This extra analysis would add some useful extra information, though is not essential.

Habitat specialisation in the heterogeneous environments provides a plausible explanation for the finding that larvae tended to specialise on one background colour over another. e.g. the larvae wouldn’t be expected to switch to an intermediate colour if they could always choose to sit on a single background type. Given the authors were unable to track individual movements I think they should present this as the most likely strategy. If the dowelling were striped (so that the larva’s bodies are always crossing many colour patches) the intermediate strategy might be more believable, however even then it could be due to a simple mechanistic constraint that only allows the larvae to sense very small patches of their substrate colour near their eyes/sensing organs.

L152-163 The definitions of colour and luminance used here are mentioned before talking about the visual system being used, and how luminance is calculated (i.e. bluetit double cone catch quanta under a given illuminant). Chroma would normally be the distance from the centre of a triangular (trichromatic) or tetrahedral (tetrachromatic) colour space, and hue would be the angle(s), but the authors seem to use chroma to refer to both hue and chroma (which is often called saturation). I think the authors should change their definitions to more standard ones, or justify why they are combining hue with chroma. To make matters more complicated chroma is later defined as ‘greenness’ in L176.

L157-158 What exactly is being being measured in these statistics? Green and brown dowels would probably differ much more in hue more than chroma.

L210 This calculation of a ‘greenness’ opponent channel has been used by others before, though using MW/(MW+LW) would generate numbers that are more biologically relevant (symmetrically bounded between zero and one, instead of asymmetric bounding between zero and infinity). The calculation of greenness from the photographs later on does this correctly. This might, for example, result in a better fit in figures 5 and S2, which shows a disproportionate effect by the most green treatment.

L216-221 How were the images linearised (the authors should cite the paper and methods used to do the objective image analysis)?

Figure 3 I think the y-axis should be labelled “larvae greenness” if this is correct. Also, the dowel measurements should be labelled and perhaps segregated more clearly from the larvae measurements.

L240-241 and 254-256 The use of an asymmetric greenness measure (as mentioned above) means that these discrepancy values cannot be compared and should probably be replaced with something more meaningful. i.e. the greenness measure is certainly not a perceptually uniform colour-space in which to look at distances. Distances in tetrahedral colour space are also somewhat problematic as these too are not perceptually uniform. Calculating JNDs would be the easiest way to quote colour differences (and would probably be useful as it is biologically meaningful).

·

Basic reporting

.

Experimental design

.

Validity of the findings

.

Additional comments

Colour change of twig-mimicking peppered moth caterpillars REVIEW

GENERAL
I am happy with this very interesting MS that has been thought out conceptually and tested carefully. Excellent literature review in Intro but seems rather duplicated in Discussion – you need only say it once, probably most of it in Discussion.

I would like a slight change of emphasis in the Introduction to explain why it is so important to explore the issue of slow colour change. It makes an attempt at this but gets lost somewhere in the prose. You may want to cut the Intro length while doing this – it is really very long at present – aim for 3 paras. For example only last 2 sentences of first para really needed! Paras 2 and 3 are really about mechanism – not what is covered in this MS.

There is remarkably little said about masquerade in the MS which is odd – after all, these organisms resemble twigs to our eyes anyway. Would we expect greater or less background matching in masquerading than non-masquerading species? Is it a synergistic trait, or a trade-off between shape and colour. Aren’t there comparable data with Kallima butterflies and striped skunks – and probably more that I don’t know of? You might want to discuss these alternatives ways of looking at the colour/shape masquerade relationship in the Intro?

It would be good to have more natural history: are these larvae sedentary on one plant – are there different parts of the plant with different colours? What are the challenges faced by individual larvae in terms of colouration?

Are there any data on this model species that pertain to larval colour and the industrial pollution story?

Figure 1 excellent


SPECIFIC
23 larval stages unclear
35 rigorously is a bit strong
168 what is a B&Q?
291 How do you know visual capabilities?
304 ref not in italics
360 any anecdotal evidence for weak selection?
Table 1 to SOM

Tim Caro

---

## Round 0.2 · accepted · Accept

· Academic Editor

Accept

Both reviewers and I agree that you've done an excellent job with the revisions. It is my pleasure to accept the paper for publication.

·

Basic reporting

basic reporting - pass

Experimental design

experimental design - pass

Validity of the findings

Validity of the findings - pass

Additional comments

The authors have been very thorough in addressing all of my concerns and queries, and I am happy to recommend acceptance. They have done a good job of acknowledging the minor limitations of their study, and making these clear to the reader.

·

Basic reporting

.

Experimental design

.

Validity of the findings

.

Additional comments

I have carefully read the revised manuscript entitled: Colour change of twig-mimicking peppered moth larvae is a continuous reaction norm that increases camouflage against avian predators. The authors have made considerable efforts to address the points raised by the editor and 2 reviewers, and I am happy with the outcome. I am not sure if the Conclusion section is needed but otherwise it is ready for publication.